# Current Status and De Novo Synthesis of Anti-Tumor Alkaloids in *Nicotiana*

**DOI:** 10.3390/metabo13050623

**Published:** 2023-04-30

**Authors:** Md. Ahsan Habib, Md. Mobinul Islam, Md. Mukul Islam, Md. Mohidul Hasan, Kwang-Hyun Baek

**Affiliations:** 1Department of Plant Pathology, Hajee Mohammad Danesh Science and Technology University, Dinajpur 5200, Bangladesh; a.habib@hstu.ac.bd (M.A.H.); mobin@hstu.ac.bd (M.M.I.); mukul_plp@hstu.ac.bd (M.M.I.); 2Department of Biotechnology, Yeungnam University, Gyeongsan 38541, Republic of Korea

**Keywords:** *Nicotiana*, alkaloids, anti-tumor, metabolic engineering, de novo synthesis

## Abstract

Alkaloids are the most diversified nitrogen-containing secondary metabolites, having antioxidant and antimicrobial properties, and are extensively used in pharmaceuticals to treat different types of cancer. Nicotiana serves as a reservoir of anti-cancer alkaloids and is also used as a model plant for the de novo synthesis of various anti-cancer molecules through genetic engineering. Up to 4% of the total dry weight of Nicotiana was found to be composed of alkaloids, where nicotine, nornicotine, anatabine, and anabasine are reported as the dominant alkaloids. Additionally, among the alkaloids present in Nicotiana, β-carboline (Harmane and Norharmane) and Kynurenines are found to show anti-tumor effects, especially in the cases of colon and breast cancers. Creating new or shunting of existing biosynthesis pathways in different species of Nicotiana resulted in de novo or increased synthesis of different anti-tumor molecules or their derivatives or precursors including Taxadiane (~22.5 µg/g), Artemisinin (~120 μg/g), Parthenolide (~2.05 ng/g), Costunolide (~60 ng/g), Etoposide (~1 mg/g), Crocin (~400 µg/g), Catharanthine (~60 ng/g), Tabersonine (~10 ng/g), Strictosidine (~0.23 mg/g), etc. Enriching the precursor pool, especially Dimethylallyl Diphosphate (DMAPP), down-regulating other bi-product pathways, compartmentalization or metabolic shunting, or organelle-specific reconstitution of the precursor pool, might trigger the enhanced accumulation of the targeted anti-cancer alkaloid in *Nicotiana*.

## 1. Introduction

Tobacco (*Nicotiana* spp.), a series of seventy-six naturally occurring species belonging to the Solanaceae family is cultivated around the world [1,2]. Several *Nicotiana* species are used for curing types of diseases and for recreation as they serve as a reservoir of a wide range of secondary metabolites viz. alkaloids, aromatic compounds, flavonoids, volatile compounds, acyclic hydroxygeranyllinalool, diterpene glycosides, etc. [3,4]. Therefore, plants play a vital role in the medicinal and agricultural industries. However, *N. tabacum* is the most popular and important species, which provides around two thousand five hundred characterized metabolites chemicals, so far [5]. Alkaloids, a major group of secondary metabolites are present in tobacco and are mostly responsible for its biological properties. Alkaloids are structured with nitrogen atoms with a ring structure, where the nitrogen atom is located inside the heterocyclic ring [6].

Tobacco plants contain 2–4% alkaloids of their total dry weight, where nicotine shares approximately 90% of total alkaloids [7]. Nornicotine, anatabine, and anabasine are the other structurally related alkaloids present in tobacco. Piperidine or pyrrolidine rings, with a positively charged nitrogen atom present in pyridine alkaloids, play the core role in toxicity against herbivores [8]. Furthermore, pyridine alkaloids and other nicotine analogs also have a toxicity effect and hence, are used to treat anxiety, different types of cancers, depression, pain, etc. [9,10,11].

Cancer, a grave threat to human beings is responsible for around ten million deaths in the world because of rare and expensive treatment [12]. Natural alkaloids including vinblastine, camptothecin, terpenoids (farnesol, geraniol, paclitaxel), anthranilic acid derivatives (tranilast), polyphenolic compounds (gossypol), lignans (podophyllotoxin), etc. commonly have the properties to act against tumor cells [5]. Tobacco plants contain a variety of alkaloids or other secondary metabolites and are considered a major source for the effective treatment of tumor cells. Cembranoidtype diterpenes (CBDs), which originated in tobacco, showed potential properties in neuroprotective functions and treating cancer cells [13]. Because of limited availability with high synthetic production costs, research around the world is focusing on the search for a new source of potential alkaloids or other strategies, which can offer an easier and less expensive means of cancer treatment. Plant suspension culture or metabolic engineering of short-cycled plants with the desired gene could offer reliable platforms for large-scale production of the targeted anti-tumor alkaloid [14]. In addition to the contribution to the fields of traditional agricultural and pharmaceutical industries, several species of tobacco including *N. benthamiana*, *N. attenuata*, etc. are considered an ideal model plant system for the production of the valuable alkaloid [15]. Tobacco can grow in a short space, has a short life cycle, can manipulate the gene easily and has high disease susceptibility [16]. Successful production of anti-tumor alkaloids or alkaloid precursors has already been achieved by the genetic engineering approaches in *Nicotiana* plants [17]. In this review, recent advances in the context of anti-cancer alkaloids and their de novo synthesis in tobacco plants are summarized. Finally, how genetic engineering could be useful for the genetic manipulation of tobacco has also been highlighted.

## 2. Plant Secondary Metabolites

Plant secondary metabolites, also known as idolizes, are chemical compounds derived from plant cells through a variety of metabolic pathways [18]. Likewise, primary metabolites and secondary metabolites do not directly participate in the growth and development of the plant. A variety of biological properties including antimicrobial, anti-tumor, etc. provides the scientific base for the use of secondary metabolites obtained from diversified herbs. To date, about fifty thousand secondary metabolites have been identified in the plants and the mode of action of many of them is yet to be explored [19]. Plant secondary metabolites are categorized into four major classes: alkaloids, phenolic compounds, sulfur-containing compounds and terpenoids [20]. However, we focus on the anti-tumor alkaloids present in or produced in *Nicotiana* through genetic engineering.

## 3. Alkaloids: The Major Plant Secondary Metabolites

Alkaloids are the more diversified nitrogen-containing compounds composed of around twenty thousand members. Among the identified alkaloids, approximately six hundred members are found to have antioxidant and antimicrobial properties, serve as a drug, have anti-cancer properties, and can stimulate the animal nervous system [21,22,23,24]. Based on molecular structure and biosynthetic pathway, alkaloids can be divided into three major groups viz. (a) true alkaloids (heterocyclic), (b) proto alkaloids (nonheterocyclic), and (c) pseudo alkaloids. The chemical composition of *Nicotiana* leaves is exceptionally complex, where nicotine is found to be the utmost distinctive member of the alkaloid family [25]. In alkaloids, a pyridine ring and a pyrrolidine ring composed the structure of nicotine, whereas a pyridine ring and a piperidine ring composed the structure of anabasine [8]. Decarboxylation of ornithine-by-ornithine decarboxylase or arginine-by-arginine decarboxylase produced putrescine, which finally served as the precursor for the derivation of the pyrrolidine ring. 4-amino butanol is produced from Nmethylation (by N-methyltransferase) and oxidatively deamination (by N-methyl putrescine oxidase) spontaneously cyclized to create 1-methyl-Δ1-pyridinium cation followed by coupling with a pyridine ring originated from nicotinic acid to synthesize nicotine. In contrast, decarboxylations of lysine (decarboxylase) produce cadaverine, the precursor for the generation of the piperidine ring for the formation of anabasine. The oxidation of cadaverine by amine oxidase and subsequent cyclization produces a Δ1-piperidine ring, which couples with the pyridine ring just in the same manner for the derivation of nicotine to synthesize anabasine (Figure 1) [8].

## 4. Alkaloid Present in *Nicotiana*

Alkaloids are the major secondary metabolites (up to 4% of DW) found in *Nicotiana* and their occurrence varies from species to species [7]. To date, around seventy-one alkaloids have been reported in *Nicotiana* spp. where, nicotine, nornicotine, anabasine, and anatabine are reported as the most abundant alkaloids (Figure 2) [26,27].

## 5. Anti-Cancer Alkaloids Present in *Nicotiana*

### 5.1. β-Carboline

β-carboline are the bioactive alkaloids naturally produced in plants, foods, cigarette smoke, and mammalian tissues as well as in the human brain exhibiting antimicrobial, neuroactive activities and used to treat cancer [28,29,30]. Harmane and norharmane belonging to β-carboline are present in Nicotiana (up to 20 μg/g), which also possess antimicrobial, anticonvulsant, neuroprotective, and anti-tumor properties [31,32]. They act as mutagenic agents by inhibiting different enzymes’ functions including histone deacetylase, receptors of the central nervous system [33]. Norharmane consists of benzene, pyrrole, and pyridine ring where C-1, C-3, and N-9 nucleus positions serve as active sites used to develop new molecules with anti-tumor activity [34,35]. Derivatives of nonharmane, namely, harmane and harmine and other norharmane salicylic conjugate amides can depolarize the mitochondria and are also used to treat liver and colon cancer [34]. Tryptophan amino acid consists of carbon skeleton and nitrogen atoms and serve as the precursors of β-carboline. Amine derivatives or indolethylamino acid undergo an enzymatic reaction (Pictet–Spengler) and can react with an aldehyde acid or a keto acid to produce a Schiff base intermediate precursor form, from which tetrahydro β-carboline is derived through a cyclization process [36]. After that, tetrahydro β-carboline (1-methyl-1,2,3,4-tetrahydro-b-carboline-3-carboxylic acid and 1,2,3,4-tetrahydro-b-carboline-3-carboxylic acid) oxidizes by heme peroxidases and produce β-carbolines [36].

### 5.2. Kynurenines

Kynurenine (6-hydroxykyrunenine) and some of its derivatives isolated from leaves of *Nicotiana tabacum* help to relax the arterial vessels, control blood pressure, and boost immunity in response to inflammation [37,38,39]. The compound is derived from tryptophan and acts as the precursor for the derivation of kynurenic acid, anthranilic acid, and 3-hydroxykynurenine [27,40]. In the course of the Kynurenine pathway, tryptophan is catalyzed by indole-2,3-dioxygenase and tryptophan-2,3-dioxygenase to generate N-formyl-kynurenine. Further, kynurenine formylase catalyzes N-formyl kynurenine to L-kynurenine. Finally, kynurenine 3-monooxygenase or kynureninase directs kynurenine aminotransferase for the conversion of kynurenine into Kynurenic acid or 3-hydroxyanthranillic acid derived [41,42,43].

### 5.3. Nicotine and Nornicotine

Nicotine is the major (0.6~3.0% of the dry weight) alkaloid present in *Nicotiana* spp. [44,45]. Nicotine does not initiate cancer but can affect cancer development through activation and binding with the acetylcholine receptors to synthase tobacco-specific N-nitrosamines or inhibition of immune response by affecting dendritic cells [46]. Nornicotine (2-pyridin-3-ylpyrrolidine-1-carbaldehyde) is a chemical analog to nicotine without methyl group (3–5% of total alkaloid) and the synthesis of carcinogen N-nitrosonornicotine during the curing and processing of tobacco acts as a precursor [47,48]. A known type 1 carcinogen N-nitrosonornicotine is formed in human saliva for its action [49]. Interestingly, estrogen biosynthesis (for cancer development) was found to reduce using aromatase inhibitors like nornicotine in the case of breast cancer-infected and smoking people. Nornicotine and anabasine acyl derivatives such as N-(4-hydroxyundecanoyl) and N-n-octanoylnornicotine also can hinder estrogen synthesis in cancer cells [50]. Though carbaldehyde compounds are effectively used for cancer treatment, nornicotine has also been claimed for therapeutic and medical purposes because of its carbaldehydne nature and potential antimicrobial, anti-inflammatory, antioxidant, and anti-cancer properties “https://www.benchchem.com/product/b014642 (accessed on 2 April 2023)” [51]. A defensin-type protein (NAD1) in the flower of Nicotiana alata also showed anti-cancer efficacy “https://theconversation.com/tobacco-plants-may-contain-cure-for-cancer-a-new-twist-in-protein-lipid-interactions-25271 (accessed on 8 April 2023)”. However, still there is a debate whether nicotine or its derivative act as an anti-cancer agent or not. Before, making any conclusion regarding the anti-tumor properties of those compounds, more in-depth research with proper proof needs to be explored [46].

## 6. Metabolic Engineering of *Nicotiana* for Anti-Cancer Compound

For ease of genetic modification and cultivation, different species of *Nicotiana* are widely used in biosynthetic pathway reconstitutions of various valuable anti-cancer alkaloids [52,53].

### 6.1. Taxol

Paclitaxel (Taxol) is a natural alkaloid that was isolated from the bark of Taxus brevifolia at a very low concentration (0.01%) and was found to be very effective to treat various malignancies like ovarian cancer, lung cancer, breast cancer, kidney failure, restenosis, rheumatoid arthritis, etc. [54,55,56]. The biosynthetic pathway of taxol has nineteen steps from GGPP (geranylgeranyl pyrophosphate) [57] including several cytochrome P450 (CYP) mediated modifications [58], hence, its enzymatic production is very high [59,60,61]. Nicotiana benthamina was used to produce taxadiene, the core skeleton of taxol through the successful introduction and integration of the taxadiene synthase gene (TS gene) [15]. The transformed *N. benthaniana* plants with TS genes containing CaMV 35S promoter leads to the de novo synthesis of taxadiene in the leaves (11–27 µg taxadiene/g dw) and the roots (14.6–22.5 µg/g) [15,62]. Along with the de novo synthesis of taxadiene, in *N. benthamina*, taxadiene-5α-ol was also produced through the compartmentalization of cytochrome P450 reductase, T5αH, and TS in the chloroplast coupled with the elicited pool of isoprenoid precursor [63]. In the genetically engineered *N. benthamiana*, silencing or shunting of the existing metabolic pathway directed by the phytoene synthase gene demonstrated a 1.4- or 1.9- fold increase in the synthesis of taxadiene [15]. Suppressing carotenoid synthesis by shunting the phytoene synthase in the TS- transformed *N. benthamiana* increased the taxadiene synthesis by 1.9-fold, whereas silencing of the phytoene desaturase gene, the second devoted step, failed to redirect the GGPP pool for increased taxadiene production because of the interference of a newly formed biosynthetic pathway or some unknown reasons (Figure 3) [15,64].

### 6.2. Artemisinin

Artemisinin, a sesquiterpene alkaloid present in the aerial parts of Artemisia annua with anti-cancer properties is effectively used in pharmaceutical industries [65,66]. Artemisinin is derived through a general terpenoid biosynthesis pathway, where farnesyl diphosphate synthase (FPPS/FPS) helps to unite isopentenyl diphosphate (IPP) with dimethylallyl diphosphate (DMAPP) for the synthesis of farnesyl diphosphate (FPP, farnesyl pyrophosphate) [67,68]. Through carbocation formation and cyclization, FPP is transformed to amorpha-4, 11-diene catalyzed by amorpha-4, 11-diene synthase (ADS), which is further hydroxylased into artemisinic alcohol and then oxidized by amorphadiene monooxygenase (CYP71AV1) to artemisinic aldehyde [69,70,71]. Artemisinic aldehyde 11(13) reductase (DBR2) further reduced artemisinic aldehyde into dihydro artemisinic aldehyde, which is then oxidized by aldehyde dehydrogenase (ALDH1) to dihydroartemisinic acid [72,73]. The accumulation of artemisinin is affected because of the transformation of dihydro artemisinic aldehyde to dihydro artemisinic alcohol incited by dihydro artemisinic aldehyde reductase (RED1) [72]. Finally, a spontaneous light-depended non-enzymatic reaction yielded artemisinin from dihydroartemisinic acid [74].

Incorporating the genes or their transient expression related in the heterologous plants yielded artemisinin [75,76,77]. *Nicotiana* spp. has also been used in artemisinin research for its availability, flexibility to accept foreign genes with swift growth and high biomass. *N. tabacum* modified with the diverse genes of MVA results in enhancing the IPP pool, which increases production of artemisinin up to 0.8 mg/g dw [75,78]. A higher accumulation of amorpha-4,11-diene, the initial product in the synthesis of artemisinin was also achieved in *Nicotiana tabacum* through the expression of ADS [79]. However, the accumulation of amorpha-4,11-diene increased up to 4 mg/g fresh weight after the simultaneous incorporation of CYP71AV1, DBR2, and ALDH1 with ADS [76]. Other than *N. tabacum*, artemisinic acid or glycosylated artemisinin precursors were also produced in *N. benthamiana* through the transient expression of ADS, HMGR, CYP71AV1, and FPS or artemisinin genes [77,80].

Introduction of the artemisinin pathway through the transformation of the plastid genome in the chloroplasts of *N. tabacum* overcame the problem and resulted in higher artemisinic acid accumulation (120 µg/g) [20,81]. The introduction of six genes from the mevalonate pathway targeting the chloroplast accompanied by artemisinin pathway genes insertion into nuclear genome of *N. tabacum* through chloroplast transit peptide produced a higher amount of artemisinin (∼0.8 mg/g dry weight) [78]. Yet the significant production of the compound is not possible because of the biosynthesis pathway and multifaceted behavior of gene expression along with the composite glycosylation process (Figure 3) [82].

### 6.3. Parthenolide

Parthenolide mostly obtained in the feverfew plant (*Tanacetum parthenium*) is a sesquiterpene lactone that serves as a drug, especially for the treatment of colon cancer [83]. Structural parthenolide biosynthetic pathway genes including germacrene A oxidase (TpGAO), germacrene A synthase (TpGAS), parthenolide synthase (TpPTS), and costunolide synthase (TpCOS) were isolated from the feverfew plant [84]. A transient heterologous gene expression of TpGAO, TpGAS, TpPTS, and TpCOS coding sequences was cloned into pBIN binary expression vector under the Rubisco promoter control and introduced into the *N. benthamiana* plants. The reconstituted pathway did not result in any free parthenolide in the leaf of transformed *N. benthamiana*, however, a minor amount of parthenolide (2.05 ng/g FW) was produced when FDP precursor supply was boosted through the addition of AtHMGR. Interestingly, some parthenolide conjugates, namely, cysteine and glutathione were also produced along with parthenolide (1.4 μg/g) (Figure 3) [85].

### 6.4. Costunolide

Costunolide is a well-known sesquiterpene lactone present in several medicinal plants including Magnolia grandiflora and Tanacetu parthenium [86]. Costunolide is used to treat different types of cancers including leukemias, breast cancer, liver cancer, etc. [87,88,89]. Transient expression with feverfew germacrene A synthase (TpGAS), chicory germacrene A oxidase (CiGAO), and chicory costunolide synthase (CiCOS) in *N. benthamiana* produce costunolide up to 60 ng/g FW. The costunolide precursor germacrene A increases with mitochondrial TpGAS steering as compared to the cytosol targeting. However, when the leaf is infiltrated with the CiGAO and TpGAS, germacrene A disappeared due to the effect of CiGAO. This happened due to the CiGAO enzyme, which converts germacrene A into germacra-1(10), 4, 11(13)-trien-12-oic acid (Figure 3) [89].

### 6.5. Etoposide and Related Anti-Cancer Molecules

Etoposide obtained from the mandrake plant (*Podophyllum peltatum*) is an alkaloid used for the treatment of gastric cancer, testicular cancer, germ cell tumors, breast cancer, Hodgkin’s and non-Hodgkin’s lymphomas as well as lung cancer by preventing DNA unwinding through the inhibition of the function of topoisomerase II [90,91]. In *N. benthamiana*, the etoposide production pathway was reprogrammed by Agrobacterium- based transient expression using a single lignin-associated transcription factor and MYB85, which resulted in increased etoposide aglycone (EA) production by two times (up to 1 mg/g, DW), deoxypodophyllotoxin (DPT), the last biosynthetic anti-cancer precursor of the etoposide aglycone (EA) production pathway by eight times (35 mg/g DW) and epipodophyllotoxin (3.5 mg/g DW) [92]. Coniferyl alcohol (CA), a monolignol produced from the L-phenylalanine in the Podophyllum spp. acted as a building block to produce lignin compounds, which also acted as the precursor for the synthesis of etoposide. Agrobacterium containing the DPT pathway genes were infiltrated along with coniferyl alcohol (CA) resulting in a thirteen-fold increase in DPT production as paralleled to no infiltration of CA. Transient expression of *N. banthamiana* with sixteen genes includes coniferyl alcohol and enzymes of the etoposide production pathway resulting in 4.3 mg/g DW DPT production in the leaves. [93]. On the other hand, agro-infiltration of eight genes of the DPT pathway without coniferyl alcohol genes into the *N. banthamiana* also resulted in increased synthesis of DPT. Along with the mentioned genes, the addition of (+) pinoresinol resulted in the eight-fold elicited production of DPT in the same heterologous plant system [94]. Increased production of DPT through genetic manipulation using various pathway-related genes including enzymes responsible for etoposide aglycone and coniferyl alcohol (CA), proved a significant way to increase etoposide production (Figure 3) [91,93].

### 6.6. Crocin

Crocin (crocetin digentiobiose ester) is the alkaloid present in saffron (*Crocus sativus*) and is used to treat cancer as it inhibits the mitotic cell division, triggering apoptosis and proliferation of cells [95,96]. The enzyme carotenoid cleavage dioxygenase 2L (CsCCD2L) plays a vital role in the crocin biosynthesis pathway. Agrobacterium-mediated genetic transformation of *N. tabacum* and *N. glauca* using the orange mutant gene of Arabidopsis thaliana (AtOrMut) and β carotene hydroxylase (BrCrtZ), CsCCD2L through Arabidopsis AtUBQ10, tobacco polyubiquitin Ubi.U4 and promoter CaMV35S with a marker of hygromycin gene, resulted in ten times increased synthesis of crocin in *N. glauca* (400 µg/g DW) as compared to *N. tabacum* (36 µg/g DW) (Figure 3) [97].

### 6.7. Vinblastine

Vinblastine is a pharmaceutical agent used to treat various types of cancer derived from Catharanthus roseus [21]. The vinblastine production pathway is composed of thirty-one enzymes from geranyl pyrophosphate, where strictosidine monoterpene indole alkaloid is used as the precursor [98]. Agrobacterium-mediated transient expression of *N. banthamiana* with six stemmadenine acetate biosynthesis genes, namely, strictosidine glucosidase (SGD), geissoschizine synthase (GS), redox1, redox2, geissoschizine oxidase (GO) and stemmadenine acetyltransferase (SAT) from *C.* reseus, was carried under the controlling of SIUbq10 promoter by using the Golden Braid assembly system along with a P19 silencing suppressor to escape RNA silencing deleterious effects [99]. Further infiltration of the infiltrated leaves with strictosidine substrate resulted in no synthesis of stemmadenine acetate rather than the synthesis of stemmadenine acetate oxidized compound, namely, precondylocarpine acetate. Further, reconstitution of catharanthiane and tabersonine pathways by co-infiltration using precondylocarpine acetate synthase (PAS), dihydroprecondylocarpine synthase (DPAS), and catharanthine synthase (CS) or tabersonine synthase (TS) genes under the transcription control of a SIUbq10 promoter demonstrated increased accumulation of the precursor of vinblastine, namely, tabersonine and catharanthine (Figure 3) [98].

### 6.8. Strictosidine

Strictosidine is the last core skeleton biosynthetic precursor, first isolated from Rhazya stricta [100,101,102]. Strictosidine is produced from the amino acid tryptophan decarboxylation product tryptamine and the monoterpene precursor loganin, through the production of secologanin [103]. A higher level of strictosidine (0.23 mg/g DW) was produced in *N. banthamiana* through the reconstituted pathway genes including GPPS (Geranyl Diphosphate Synthase) and MLPL (Major Latex Protein-like enzyme) [17]. In the previous concept to maximize the synthesis, a thirteen step biosynthesis pathway needed to be reprogrammed following two phases where the second phase was considered for the synthesis of an intermediate substrate (iridotrial) [104]. Co-expression of 8-hydroxygeraniol oxidoreductase (CrGOR), geraniol 8-oxidase (CrG8H) and iridoid synthase (CrISY) resulting in elicited accumulation of nepetalactol, which directly facilitates the production of a higher level of strictosidine without adding any metabolite intermediates or precursors [17]. However, major latex protein-like enzyme (MLPL) from Nepeta (catmint) with an early step in chloroplast and subsequent steps in cytosol play a crucial role in the maximum production of strictosidine in *N. benthamiana* (Figure 3) [17].

## 7. Challenges and Future Prospects

*Nicotiana* contains approximately seventy-five species made up of twelve chromosomes, as in the majority of Solanaceae crops. In the last couple of decades, the plant became one of the key platforms for the genetic engineering program with novel biological achievements including tissue culture, hybridization, genetic transformation, transient expression, gene silencing, etc. Nicotine and other minor alkaloids present in tobacco possess pharmacological properties due to their binding ability with various nicotinic acetylcholine receptors.

In addition to the naturally occurring anti-cancer alkaloids in *Nicotiana*, the plant is also used as a heterologous system for the de novo synthesis of a variety of anti-tumor alkaloids through Agrobacterium-mediated gene transfer, agroinfiltration, virus-mediated overexpression/gene silencing, and gene editing, etc. However, the de novo synthesis of various anti-tumor alkaloids, including terpenoids, is not explored at the desired level in tobacco. MEP and MVA, the two complex pathways along with a wide range of native enzymes make tobacco plants somewhat difficult to maximize de novo synthesis of the targeted compounds or their precursors. The most noticeable and reported limitation in the metabolic engineering of this plant is the supply of substrate. The limitations of the maximum supply of the substrate could be achieved through the compartmentalization of the precursor pool by working with related enzymes or engineering of flux for the formation of precursor or organelle-specific reconstitution of the precursor-synthesizing pathway [75,105]. In recent times, metabolic engineering using compartmentalization tactics has come into view as a capable and fruitful approach to overcome those limitations and has been found to increase the synthesis of terpenoids [78,81]. Silent metabolism directed by glycosylation and methylation is also a potential limiting factor in the tobacco plant for the overexpression of genes. Targeting of subcellular compartments like cytosol, mitochondria, and plastid makes possible the reconstructed pathway for a full swing mood [106]. For the elicited production of alkaloids, virus-induced gene silencing from the endogenous precursors through compartmentation of the final product can play a vital role [107,108]. Moreover, organelle-specific engineering for the metabolic catabolism of the precursor pools could be another way to boost the increased accumulation of the desired compounds. Further, metabolic flux analysis techniques, along with improved analytical abilities, optimize the carbon flux localization in metabolic networks, revealing the metabolic channel and exploring the persistent non-functional metabolic pools [109]. The tools will ultimately boost the prospective of metabolic modeling, which directs the improvement of not only the models, but also make it possible to predict the accomplishment of future genetic engineering strategies. Successful application of CRISPR/Cas9 will further help the synthesis of recombinant protein as double-knockout genes, namely dicer-like proteins 2 and 4 that accumulate maximum human fibroblast growth factor in *N. benthamiana* [110].

## 8. Conclusions

Alkaloids are basic nitrogen-containing plant secondary metabolites that naturally occur in a wide range of plants. Alkaloids containing plants including the *Nicotiana* species have been used since ancient times for therapeutic and recreational purposes [111]. Nicotine, anatabine, anabasine, and nornicotine are the predominant alkaloids present in tobacco where nicotine contains more than 90% of the total alkaloids pool [112]. The root is the primary synthesizing area of nicotine, the related alkaloids and reached leaves through the xylem vessel [113,114]. Amino acids play the role of a precursor for the synthesis of most alkaloids, which comprised the pyridine ring and pyrrolidine ring pathways [8]. Natural secondary metabolites, including alkaloids, are synthesized in the plant like tobacco, in trace amounts, and are broadly used to treat different types of cancers [115]. Different species of *Nicotiana* such as *N. benthamiana*, *N. tabacum*, and *N. glauca* are widely used as a potential platform for the de novo synthesis of various anti-cancer alkaloids, including strictosidine, vinblastine, crocin, etoposide, costunolide, parthenolide, artemisinin, etc. However, the demand for plant-derived anti-cancer alkaloids is increasing day by day. High versatility in the metabolic pathway and the ability to produce high biomass in a short time turned the tobacco plant into a potential chassis for the production of plant secondary metabolites, including anti-cancer alkaloids. Among different species of the tobacco plant, there can be easy replication of transient expression vector established *N. benthamiana* for the in vitro synthesis of small molecules and recombinant proteins, where *N. tabacum* is considered best for the in vivo or large-scale field production of the desired molecules [52,53]. The docility in nuclear and plastid transformation of *Nicotiana* plants also plays a role in its extensive use in the field of classical transgenesis-based metabolic engineering. However, the genetic makeup of the plant makes it easy for agroinfiltration, virus-induced gene silencing, or gene editing to reconstitute the synthesis pathway of endogenous metabolites and the de novo synthesis of distant metabolites. Therefore, a huge prospect prevailed for the commercial-based production of valuable alkaloids from the existing or enriched precursor’s pools, as different valued anti-cancer alkaloids have already been produced in the plant [15,75,116,117,118]. In addition, metabolic catabolism of the precursor pools with simultaneous organelle-specific genetic engineering may perhaps facilitate maximizing the yield.

## Figures and Tables

**Figure 1 metabolites-13-00623-f001:**
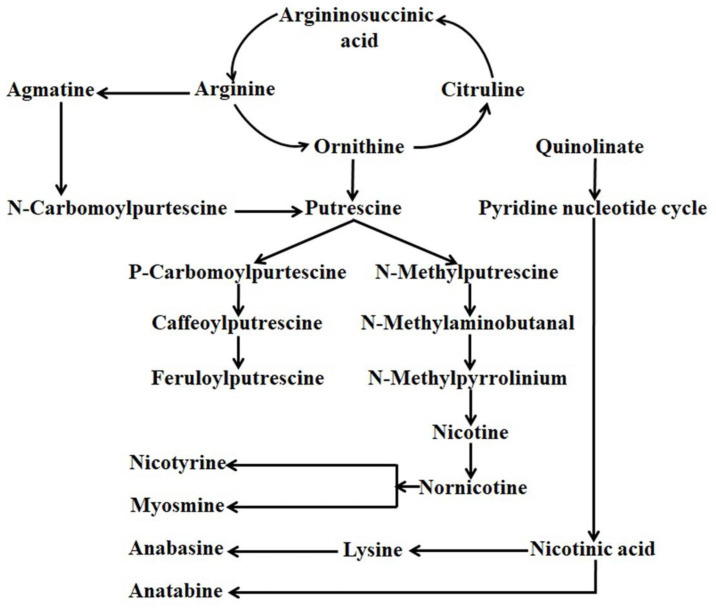
General alkaloids bio-synthesis pathway in *Nicotiana*.

**Figure 2 metabolites-13-00623-f002:**
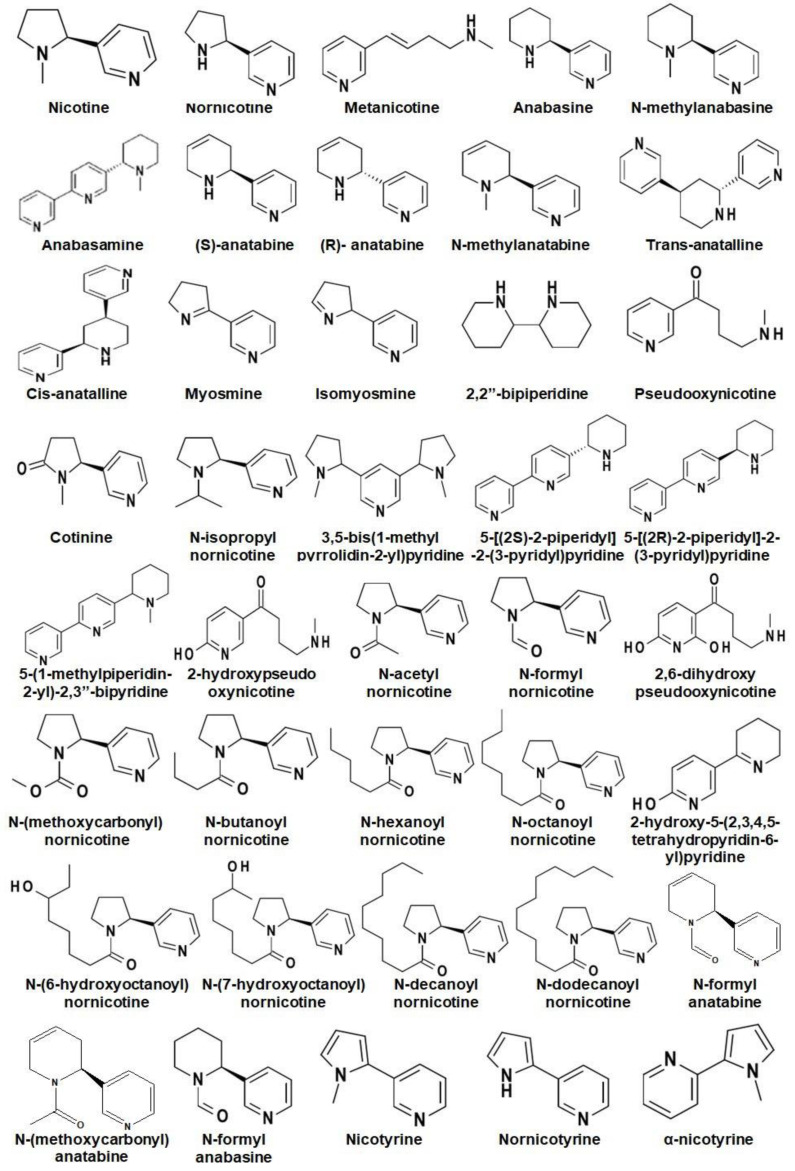
Chemical structures of different alkaloids reported in *Nicotiana*.

**Figure 3 metabolites-13-00623-f003:**
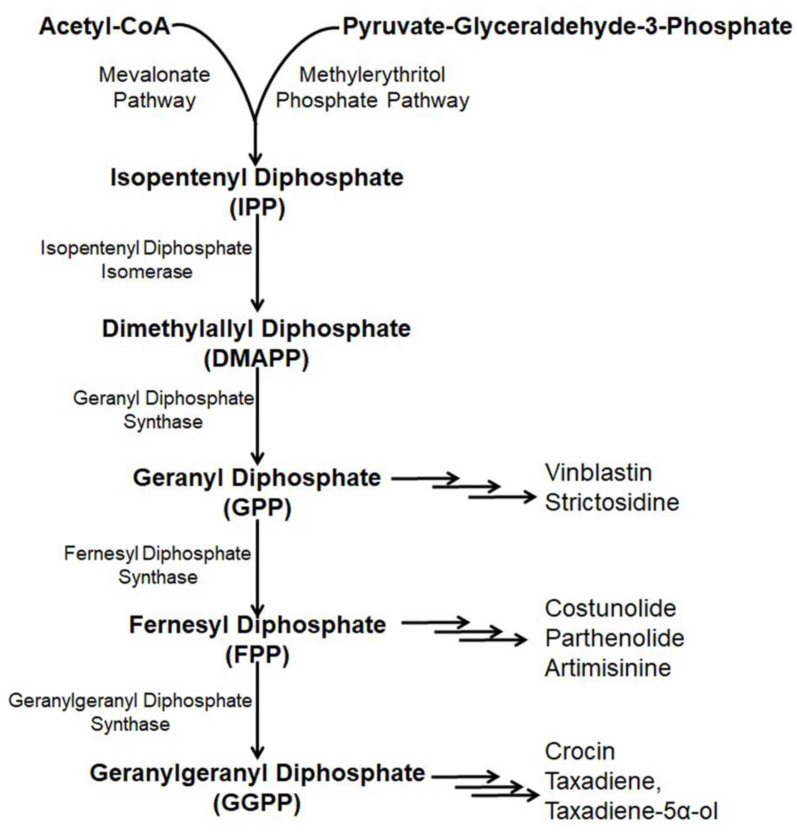
Existing and/or genetically engineered bio-synthesis pathway of different anti-cancer alkaloids in *Nicotiana*.

## Data Availability

Data is presented within the manuscript.

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
