# Peer review of "Current Status and De Novo Synthesis of Anti-Tumor Alkaloids in Nicotiana"

_metabolites, 2023, doi:10.3390/metabo13050623_

Round 1
Reviewer 1 Report
Authors reviewed the status and de novo synthesis of anticancer alkaloids in tobacco and summarized metabolic engineering of tobacco plants for anticancer compound. This review is a novel study and they highlighted the importance of anticancer alkaloids in Nicotiana.
Author Response
Dear reviewer, Thanks a lot for your appreciation about our work. Based on your suggestions, the manuscript has been thoroughly revised for other minor or major typographical or grammatical errors. Kindly see the green colors.
Reviewer 2 Report
The review “Current status and de novo synthesis of anti-tumor alkaloids in 2 Nicotiana: A mini-review” proposes a valuable, comprehensive description on the advances in the production of anticancer alkaloids and their de-novo synthesis in tobacco plants are summarized. Appreciable are the clear organization of the text and the particular attention to the potential of genetic engineering for the genetic manipulation of tobacco plants to make them to produce the desired biomolecules. Only English needs to be carefully revised, please see only few examples, to make the review suitable for publication.
Minor remarks
Line 140: “potential biological properties for blood vessel dilution and..” please make clearer
Line 153: “inhibit the immune against cancer” please revise
Line 178: “is a natural alkaloid was firstly obtained from Taxus” please correct
Lines 197-198: “second steps might not enough capable of the direction of more G” please correct
Line 395: “not only allows the chemical fingerprinting,..” please make clearer
The review “Current status and de novo synthesis of anti-tumor alkaloids in 2 Nicotiana: A mini-review” proposes a valuable, comprehensive description on the advances in the production of anticancer alkaloids and their de-novo synthesis in tobacco plants are summarized. Appreciable are the clear organization of the text and the particular attention to the potential of genetic engineering for the genetic manipulation of tobacco plants to make them to produce the desired biomolecules. Only English needs to be carefully revised, please see only few examples, to make the review suitable for publication.
Minor remarks
Line 140: “potential biological properties for blood vessel dilution and..” please make it clearer
Line 153: “inhibit the immune against cancer” please revise
Line 178: “is a natural alkaloid was firstly obtained from Taxus” please correct
Lines 197-198: “second steps might not enough capable of the direction of more G” please correct
Line 395: “not only allows the chemical fingerprinting,..” please make it clearer
Author Response
Dear reviewer, Thanks a lot for your time and suggestions. We have tried to improve the manuscript according to your comments and suggestions. The manuscript has also been thoroughly revised for other minor or major typographical or grammatical errors.
Comment 1: Line 140. “potential biological properties for blood vessel dilution and...”
Answer: The necessary correction has been done and incorporated in the manuscript marking with red color. Kindly see line 139-141.
Comment 2: Line 153. “inhibit the immune against cancer”
Answer: The necessary correction has been done and incorporated in the manuscript marking with red color. Kindly see line 151-153.
Comment 3: Line 178. “is a natural alkaloid was firstly obtained from Taxus”
Answer: The necessary correction has been done and incorporated in the manuscript marking with red color. Kindly see line 178-181.
Comment 4: Line 197-198. “second steps might not enough capable of the direction of more G”
Answer: The necessary correction has been done and incorporated in the manuscript marking with red color. Kindly see line 193-198.
Comment 5: Line 395. “not only allows the chemical fingerprinting”
Answer: The necessary correction has been done and incorporated in the manuscript marking with red color. Kindly see line 393-396.
The manuscript has also been thoroughly revised for other minor or major typographical or grammatical errors. Kindly see the green colors.